Phenotypic spectrum of FGF10-related disorders: a systematic review

Bzdega Katarzyna
Karolak Justyna A. jkarolak@ump.edu.pl
1 Chair and Department of Genetics and Pharmaceutical Microbiology, Poznan University of Medical Sciences , Poznan , Poland
Uversky Vladimir
Electronic publication date: 2022 Sep 14
Publication date: 2022
Volume: 10
Electronic Location ID: e14003
Received 2022 Jun 3; Accepted 2022 Aug 13
Copyright: ©2022 Bzdega and Karolak
Copyright year: 2022
Copyright holder: Bzdega and Karolak
License: This is an open access article distributed under the terms of the Creative Commons Attribution License, which permits unrestricted use, distribution, reproduction and adaptation in any medium and for any purpose provided that it is properly attributed. For attribution, the original author(s), title, publication source (PeerJ) and either DOI or URL of the article must be cited.
License URL: https://creativecommons.org/licenses/by/4.0/

Keywords: Pulmonary diseases, Congenital diseases, FGF10 deficiency

Funding: National Science Centre in Poland 2019/35/D/NZ5/02896 This work was supported by the grant awarded by the National Science Centre in Poland 2019/35/D/NZ5/02896 (Justyna A. Karolak). The funders had no role in study design, data collection and analysis, decision to publish, or preparation of the manuscript.

==============================
FGF10, as an FGFR2b-specific ligand, plays a crucial role during cell proliferation, multi-organ development, and tissue injury repair. The developmental importance of FGF10 has been emphasized by the identification of FGF10 abnormalities in human congenital disorders affecting different organs and systems. Single-nucleotide variants in FGF10 or FGF10-involving copy-number variant deletions have been reported in families with lacrimo-auriculo-dento-digital syndrome, aplasia of the lacrimal and salivary glands, or lethal lung developmental disorders. Abnormalities involving FGF10 have also been implicated in cleft lip and palate, myopia, or congenital heart disease. However, the exact developmental role of FGF10 and large phenotypic heterogeneity associated with FGF10 disruption remain incompletely understood. Here, we review human and animal studies and summarize the data on FGF10 mechanism of action, expression, multi-organ function, as well as its variants and their usefulness for clinicians and researchers.

Introduction

The fibroblast growth factor (FGF) signaling pathway plays an essential role in mammalian embryo formation and is crucial for mesenchymal-epithelial communication, contributing to the development of many different organs (Itoh, 2016). FGF signaling is also involved in maintaining biological homeostasis due to the regulation of metabolism and endocrine secretion (Hui et al., 2018). Disruption of the proper functioning of the FGF pathway can cause congenital disorders, metabolic diseases, or cancers (Ornitz & Itoh, 2015; Itoh, 2016).

One of the most important FGF ligands is fibroblast growth factor 10 (FGF10), which plays an essential role in airway branching (Jones, Chong & Bellusci, 2020), functioning of the cardiovascular system (Itoh et al., 2016), and development of the kidney, cecum, thymus, trachea, prostate, mammary gland, adipose tissue, and limb (Itoh, 2016). Pathogenic variants in FGF10 can cause craniofacial defects (Prochazkova et al., 2018), lung disorders (Vincent et al., 2019), myopia (Jiang et al., 2019), limb (Ohuchi et al., 2000) or genitourinary system anomalies (Milunsky et al., 2006), and heart defects (Itoh et al., 2016). Alterations in FGF10 have been associated with cancers, including breast or pancreatic cancer (Dankova et al., 2017; Ndlovu et al., 2018). Complete understanding of the developmental function of FGF10 and how its disruption influences the phenotype is an important but challenging task. Here, we systemize the current knowledge on FGF10, relevant for clinically- and research-focused scientists. We provide an insight into the significance of FGF10 in development by discussing its tissue-specific expression and related phenotypic spectrum in humans.

Survey Methodology

All manuscripts cited in this review were found and analyzed from the PubMed database (https://pubmed.ncbi.nlm.nih.gov/) using the following keywords: FGF family, FGF10 mutations, FGF10 diseases, FGF10 cancers, FGF10 expression, FGF10 role. Articles unrelated to FGF10-linked disorders in humans and mice were excluded.

FGF family and their mechanism of action

The FGF protein family consists of 22 members that, based on the differences in their biochemical nature, amino acid sequence, or evolutionary origin, are divided into seven subfamilies: FGF1, FGF4, FGF7, FGF8, FGF9, FGF11, and FGF19 (Itoh & Ornitz, 2008). Based on their mechanism of action, FGF subfamilies can be further classified as canonical, hormone-like, or intracellular FGFs (Itoh, 2010).

The activation of canonical FGFs depends on their binding to FGF receptors (FGFRs), mediated by heparan sulfate glycosaminoglycan (HSPG), leading to the formation of a ternary FGF-FGFR-HSPG complex on a cell surface (Esko & Selleck, 2002; Lindahl & Li, 2009). In contrast, hormone-like FGFs have a poor affinity for HSPG and require other co-receptors, Klotho proteins, for FGFR activation (Zhang et al., 2015; Prudovsky, 2021).

Canonical FGFs

Canonical FGFs constitute the largest group of FGFs, consisting of FGF1, FGF4, FGF7, FGF8, and FGF9 subfamilies (Giacomini et al., 2021). All but the FGF9 subfamilies are characterized by a paracrine/autocrine secretion mechanism (Imamura, 2014) and are crucial for the development of many organs, including glands and limbs (Zinkle & Mohammadi, 2019). In contrast, members of the FGF9 subfamily act only as paracrine factors and play an essential role in cardiac development and maintenance of cardiac homeostasis (Wang et al., 2018a; Khosravi et al., 2022).

Hormone-like FGFs

Members of the FGF19 subfamily belong to hormone-like FGFs that act as endocrine hormones (Dolegowska et al., 2019). The FGF19 family regulates glucose and lipid metabolism (Beenken & Mohammadi, 2009) by increasing hepatic glycogen synthesis, glucose tolerance, and insulin sensitivity and decreasing gluconeogenesis and hepatic triglyceride content (Fu et al., 2004). It is also essential for the enterohepatic circulation of bile or phosphorus and vitamin D3 homeostasis (Imamura, 2014; Dolegowska et al., 2019).

Intracellular FGFs

Intracellular FGFs from the FGF11 subfamily demonstrate FGFR-independent intracrine activity (Dolegowska et al., 2019). Members of the FGF11 subfamily are involved in neuronal development (Wiedlocha, Haugsten & Zakrzewska, 2021). They regulate voltage-gated sodium channel activity in neurons and are required for axon development, neuronal migration in the cerebral cortex, and microtubules stabilization (Wu et al., 2012; Zhang et al., 2012).

Activation of the FGF10 binding

FGFRs are high-affinity trans-membrane surface tyrosine kinase receptors encoded in humans by the FGFR1, FGFR2, FGFR3, FGFR4, and FGFRL1 genes (Katoh & Nakagama, 2014). FGF10 binds specifically to epithelial FGFR2b, characterized by three domains, extracellular, transmembrane, and intracellular tyrosine kinase (Itoh, 2016; Xie et al., 2020). In the extracellular domain of FGFR2b, three immunoglobulin-like subdomains (I, II, and III) are distinguished, where II and III subdomains represent the sites of HSPG and FGF10 ligand binding (Fig. 1) (Itoh, 2016; Ferguson, Smith & Francavilla, 2021).

Figure 1 A simplified scheme of the FGF10/FGFR2b activation.

Numbers I-III indicate immunoglobulin-like subdomains of the FGFR2b extracellular domain, with the FGF10 ligand and HSPG binding sites between subdomains II and III. In the tyrosine kinase domains, the purple square box indicates the phosphorylation sites required for activation of the FGF receptor substrate 2 α (FRS2 α), growth factor receptor-bound protein 2 (GRB2) and phospholipase C gamma 1 (PLC γ1). After phosphorylation of PLC γ1, calcium is released and protein kinase C (PKC) is enabled. Activated FRS2 α and GRB2 lead to activation of RAS-MAPK and PI3K-AKT signaling pathways.

The FGF10 ligand binds to the FGFR2 through HSPG to initiate signaling (Watson & Francavilla, 2018). Ligand–receptor binding triggers dimerization of FGFR2b and phosphorylation of tyrosine residues within its intracellular domains, resulting in the FGFR2b activation (Belov & Mohammadi, 2013; Watson & Francavilla, 2018). Phosphorylated FGFR2b activates FGF receptor substrate 2 α (FRS2 α) and phospholipase C gamma 1 (PLC γ1) (Itoh, 2016; Watson & Francavilla, 2018). Whereas the activated FRS2α enables the recruitment of growth factor receptor-bound protein 2 (GRB2) (Ong et al., 2000; Watson & Francavilla, 2018), leading to the activation of Ras/Raf/mitogen-activated protein kinases (MAPKs) and phosphatidylinositol-3 kinase/protein kinase B (PI3K/AKT) (Ornitz & Itoh, 2015; Dai et al., 2022), PLC γ1 leads to the intracellular calcium release followed by the activation of protein kinase C (PKC) (Marchese et al., 2001; Itoh, 2016; Watson & Francavilla, 2018) (Fig. 1).

Modification of FGF10 signaling occurs through molecular cross-talk between FGF10/FGFR2b and Wnt, retinoic acid, or transforming growth factor β signaling pathways (Xie et al., 2020). Furthermore, the FGF10/FGFR2b pathway can be also regulated by several proteins that are co-expressed with FGF and can either inhibit or enhance the signaling (Fürthauer et al., 2001; Zhao & Zhang, 2001; Tsang et al., 2002; Watson & Francavilla, 2018; Chanda et al., 2019; Böttcher et al., 2004; Xie et al., 2020).

FGF10 expression

In a murine model, high Fgf10 expression was observed in the neural crest-derived mesenchyme of the forming salivary gland, mesenchyme of the developing epithelial bud of the lacrimal gland, and Harder’s gland (Makarenkova et al., 2000; Govindarajan et al., 2000; Teshima, Lourenco & Tucker, 2016).

FGF10 is expressed throughout the human lung parenchyma and partially in airway smooth muscle cells or vasculature from 10 to 21 weeks of gestation (Al Alam et al., 2015; Danopoulos et al., 2019). A constant level of FGF10 transcripts is persistent during the pseudoglandular phase of lung development and increases during the canalicular phase (Al Alam et al., 2015; Danopoulos et al., 2019). The spatial lung Fgf10 expression shows different patterns depending on the branching phases (Hirashima, Iwasa & Morishita, 2009). In the earliest stage of lung branching, elongation mode, Fgf10 expression occurs at a single site in the bud apex region in the axial direction of elongation (Hirashima, Iwasa & Morishita, 2009). During the terminal bifurcation mode, Fgf10 expression localizes into two sites between the bud apex and the lung border, while in the last stage, lateral budding, it appears at sites on either side of the stem where potential budding is located (Hirashima, Iwasa & Morishita, 2009).

FGF10 expression has also been observed in other organs, including human and mouse sclera (Lim et al., 2012) or developing human posterior colon and rectum (Yin et al., 2013). In addition, FGF10 RNA is considered an endogenous marker in the second heart field (Kelly, Brown & Buckingham, 2001).

Role of the FGF10 gene in human morphogenesis

The human FGF10 gene spanning 89 kb consists of three coding exons and maps to the reverse strand of the 5p12 chromosome (Emoto et al., 1997). The FGF10 protein has 23.4 kDa composed of 208 amino acids and comprises a signal peptide (1-37 aa) and an FGF domain (38–208 aa) with two known glycosylation sites (51 and 196 aa) (Fig. 2) (Beenken & Mohammadi, 2009). The secreted FGF10 protein proceeds through the canonical endoplasmic reticulum-Golgi secretory pathway, after which the signal peptide is cleaved (Watson & Francavilla, 2018).

The role of FGF10 begins in the gastrulation stage (Thiery et al., 2009). FGF10 is involved in the formation of lacrimal and salivary glands at the embryonic period (Prochazkova et al., 2018; Karasawa et al., 2022) by regulating the progenitor cell population and stimulating gland outgrowth (Chatzeli, Gaete & Tucker, 2017). In adults, FGF10 is critical in maintaining gland homeostasis and/or regeneration (Mauduit et al., 2022).

The FGF10/FGFR2b signaling is also required for lung development and is involved in the induction of several genes responsible for the branching of lung epithelial tubules and alveolar formation (Bellusci et al., 1997; Wang et al., 2018b; Yuan et al., 2018; Yang et al., 2021). In vitro studies have also indicated the involvement of the FGF10 signaling in fluid secretion (Graeff, Wang & McCray, 1999). It also promotes the growth of lung buds (Hines & Sun, 2014). The FGF10 gene initiates lung bud proliferation through mesenchymal-epithelial signaling via the B isoform of FGFR2 (Bellusci et al., 1997; Abler, Mansour & Sun, 2009).

Figure 2 Schematic representation of the FGF10 protein.

Blue, grey, and dark green boxes indicate signal peptide, the FGF10 domain, and glycolysation sites, respectively. Variants identified in patients with lethal lung developmental disorders (LLDD), lacrimo-auriculo-dento-digital syndrome (LADD), aplasia of the lacrimal and salivary glands (ALSG), and risk of chronic obstructive pulmonary disease (COPD) are shown as lollipops and are represented in orange, light green, dark blue, and yellow, respectively. The number of circles in the lollipop represent a number of individuals with a particular variant and two-colored circles indicate patients affected by two different conditions.

During facial formation, Fgf10 is involved in the development of the palatal shelves, mandible, or teeth (Ohuchi et al., 2000; Prochazkova et al., 2018). It is also important in cranial development and eyelid (Prochazkova et al., 2018) or lens formation (Chaffee et al., 2016). FGF10 is essential for the proliferation of hepatoblasts (Berg et al., 2007), regulation of radial glial cell differentiation, controlling the number of progenitor cells and neurons (Sahara & O’Leary, 2009), adipogenesis (Sakaue et al., 2002), and heart repair mechanisms (Rochais et al., 2014).

Role of FGF10 in diseases

Pathogenic variants in FGF10 can lead to congenital disorders involving the respiratory system as well as the lacrimal and salivary glands (Fig. 2, Table 1) (Vincent et al., 2019). In addition, a number of single-nucleotide variants (SNVs) within FGF10 have been associated with the increased risk of nonsyndromic cleft lip with or without cleft palate (NSCL/P) (Yu et al., 2017), chronic obstructive pulmonary disease (COPD) (Klar et al., 2011; Sun et al., 2021) and myopia (Jiang et al., 2021) (Table 2).

Lacrimo-auriculo-dento-digital syndrome and aplasia of lacrimal and salivary glands

Lacrimo-auriculo-dento-digital syndrome (LADD; MIM#149730) and aplasia of the lacrimal and salivary glands (ALSG; MIM #180920) are rare genetic diseases manifesting with variable expression and inherited in an autosomal dominant manner (Milunsky et al., 2006; Ryu et al., 2020). LADD and ALSG belong to the same phenotypic spectrum; however, LADD patients present a more severe phenotype than individuals with ALSG (Milunsky et al., 2006; Rohmann et al., 2006). In addition to characteristic features observed in both LADD and ALSG patients, including dryness and infections of the eyes and mouth as well as dental caries due to atresia or hypoplasia of the lacrimal and salivary glands (Milunsky et al., 2006; Seymen et al., 2017), patients affected with LADD syndrome often present with facial dysmorphism with ear underdevelopment and hearing loss, anomalies of fingers and toes, kidneys, gastrointestinal symptoms, or respiratory disorders (Ryu et al., 2020). Interestingly, both LADD syndrome and ALSG have a large variability of associated symptoms, even when patients are from the same family and have the same genetic background, making phenotype-genotype correlation challenging (Shams et al., 2007).

LADD syndrome and ALSG are caused by heterozygous SNVs or copy-number variant (CNV) deletions involving the FGF10 gene (Rohmann et al., 2006; Shams et al., 2007). LADD can also be associated with changes affecting FGFRs, FGFR2 and FGFR3 (Rohmann et al., 2006; Shams et al., 2007). In contrast to gain-of-function FGFR2 variants associated with craniosynostosis, LADD-related variants usually lead to a decrease in the activity of FGF10 signaling (Shams et al., 2007).

Table 1 List of the coding variants within FGF10 identified in patients with lethal lung developmental disorders, lacrimo-auriculo-dento-digital syndrome, and aplasia of the lacrimal and salivary glands.

Genetic findings	Diseases	References	
c.577C>T	ALSG, CAD, risk of COPD	Entesarian et al. (2005), Karolak et al. (2019) and Klar et al. (2011)	
c.467T>G	LADD	Milunsky et al. (2006)	
c.409A>T	ALSG, LADD	Milunsky et al. (2006)	
c.240A>C	ALSG	Entesarian et al. (2007)	
c.413G>A	ALSG, LADD	Entesarian et al. (2007)	
c.317G>T	LADD	Rohmann et al. (2006)	
c.237G>A,	ALSG	Seymen et al. (2017)	
c.526 del	LADD, AcDys	Karolak et al. (2019)	
c.218T>G	ALSG	Rodrigo et al. (2018)	
c.68_70del	Risk of CTD	Zhou et al. (2020)	
Notes.

ALSG aplasia of the lacrimal and salivary glands

LADD lacrimo-auriculo-dento-digital syndrome

AcDys acinar dysplasia

CAD congenital alveolar dysplasia

COPD chronic obstructive pulmonary disease

CTD conotruncal defects

Table 2 List of single nucleotide variants identified in FGF10 associated with the risk of nonsyndromic cleft lip with or without cleft palate, chronic obstructive pulmonary disease and myopia.

Genetic findings	Disease	References	
rs2973644	risk of COPD	Ren et al. (2013)	
rs1011814	risk of COPD	Ren et al. (2013)	
rs980510	risk of COPD	Smith et al. (2018)	
rs10512844	risk of COPD	Smith et al. (2018)	
rs10462065	NSCL/P	Yu et al. (2017)	
rs10473352	risk of COPD	Ren et al. (2013)	
rs339501	Risk of extreme/high myopia	Hsi et al. (2013)	
rs12517396	Risk of extreme/high myopia	Jiang et al. (2019)	
rs10941679	Risk of breast cancer	Stacey et al. (2008)	
Notes.

COPD chronic obstructive pulmonary disease

NSCL/P nonsyndromic cleft lip with or without cleft palate

An ALSG-associated variant c.577C>T (p.Arg193*) in FGF10 was first described in 2005 (Entesarian et al., 2005). A missense variant c.467T>G (p.Ile156Arg) and a heterozygous variant c.409A>T (p.Lys137*) in FGF10 were described in two unrelated LADD patients (Milunsky et al., 2006). In one proband, the pathogenic variant was inherited from her mother with ALSG (Milunsky et al., 2006). A year later, the c.240A>C (p.Arg80Ser) variant was identified in a son and father, both affected by ALSG (Entesarian et al., 2007). A missense de novo variant c.413G>A (p.Gly138Glu) was also detected in patient with ALSG associated with anomalies in the genitourinary system and coronal hypospadias, demonstrating clinical overlap between ALSG and LADD syndrome (Entesarian et al., 2007). The c.218T>G (p.Leu73Arg) variant in FGF10 was reported in three individuals with ALSG from the same family (Rodrigo et al., 2018).

A 53 kb deletion removing exons 2 and 3 of FGF10 was identified in two families with ALSG (Entesarian et al., 2005). Recently, a novel heterozygous 12,158 bp deletion involving the last two exons of the FGF10 gene was identified in a large family with members affected by LADD or pulmonary hypoplasia (Wade et al., 2021).

Based on the molecular findings and association of FGF10 with LADD and ALSG phenotypes it has been postulated that ALSG is a milder form of LADD and not a separate disease entity (Seymen et al., 2017). It is also possible that variable expressivity of FGF10 variants results from other cis or trans genetic variants (Milunsky et al., 2006).

Phenotypes of mice with Fgf10+/− deficiency partially recapitulate the phenotypes observed in humans with FGF10 abnormalities (Entesarian et al., 2005). For example, adult Fgf10+/− mice have shown aplasia of lacrimal glands and hypoplasia of salivary glands, similar to the characteristic features observed in ALSG patients (Entesarian et al., 2005).

Lethal lung developmental disorders (LLDD)

Lethal lung developmental disorders (LLDDs) are rare diseases of newborns, characterized by severe respiratory failure, refractory to treatment (Vincent et al., 2019). LLDDs include alveolar capillary dysplasia with misalignment of pulmonary veins (ACDMPV) that is mainly caused by changes within the FOXF1 locus, and acinar dysplasia (AcDys), congenital alveolar dysplasia (CAD), or primary pulmonary hypoplasia (PH) (Vincent et al., 2019). Recent studies have shown that 45% and 20% of AcDys, CAD, or PH is associated with heterozygous SNVs or CNVs involving TBX4 and FGF10, respectively (Vincent et al., 2019). Moreover, a homozygous variant c.764G>A (p.Arg255Gln) in FGFR2 was reported in a neonate with AcDys accompanied by ectrodactyly and inherited from consanguineous parents (Barnett et al., 2016). A heterozygous FGF10 variant c.526del (p.Met176Cysfs*5) has been identified in a patient with AcDys (Karolak et al., 2019). Interestingly, the variant c.577C>T (p.Arg193*), previously detected in a family with ALSG syndrome (Entesarian et al., 2005), was also found in a patient with CAD (Karolak et al., 2019). Additionally, in two unrelated families, patients with severe lethal PH were found to have a heterozygous deletion in 5p12 (∼2.18 Mb and ∼2.32 Mb in size), involving FGF10, inherited from their parents with LADD syndrome who had no evidence of lung disease (Karolak et al., 2019). A similar phenomenon was observed in a Dutch family, in which a proband with PH inherited a heterozygous deletion involving the FGF10 gene from a mother with LADD syndrome (Wade et al., 2021).

The observation of various phenotypes in a single family with the same FGF10 variant suggests a complex model of inheritance (Karolak et al., 2019). It was postulated that haploinsufficiency of FGF10 alone is not sufficient to induce AcDys or CAD, but requires additional genetic modifiers, such as, e.g., non-coding variants (inherited or de novo; rare or common) in regulatory elements (Karolak et al., 2019). Interestingly, in the study described by Karolak et al., all infants with lung disease and coding variants in the FGF10 gene also had at least one non-coding SNV within the lung-specific enhancer ∼70 kb upstream of TBX4 (Karolak et al., 2019).

The lung phenotypes observed in patients with mutations in FGF10 were similar to that observed in mice with FGF10 abnormalities. Homozygous knockout of Fgf10 leads to a severe lung phenotype in mice and to death shortly after birth due to impaired lung morphogenesis (Min et al., 1998; Sekine et al., 1999). Mice without functional Fgfr2 die around implantation, whereas mice deficient in Fgfr2 isoform IIIb survive until birth but die shortly after due to the lack of lungs (Arman et al., 1998; Arman et al., 1999; De Moerlooze et al., 2000). The same lethal phenotype observed in Fgf10−/− and Fgfr2b−/− indicates that FGF10 is active as a specific ligand for FGFR2b (Ohuchi et al., 2000).

Chronic obstructive pulmonary disease (COPD)

COPD is characterized by abnormalities in the lung epithelium and airspace (Klar et al., 2011), which contributes to irreversible and progressive changes in airflow with an impaired response to pathogens (Tashkin et al., 1996; Rabe et al., 2007). About 26% of individuals with defects in airway development have been found to have an increased risk of COPD (Prince, 2018). Therefore, genetic factors may be important in the etiology of this disorder (Molfino, 2004).

A study of two Swedish ALSG/LADD families, including twelve affected members with the heterozygous 53 kb deletion (n = 10) or c.577C>T (p.Arg193*) variant (n = 2) involving FGF10 (Entesarian et al., 2005), showed that haploinsufficiency of FGF10 is associated with compromised lung function and likely a risk factor for COPD (Klar et al., 2011). Another study performed in the Han Chinese population of North China revealed the apparent association of the rs2973644 and rs10473352 variants in FGF10 with COPD, and suggested that rs1011814 might be responsible for the severity of COPD (Ren et al., 2013). Additionally, recent analyses showed that FGF10 variants rs980510 and rs10512844 are associated with the absence of the proper medial-basal airway in smokers, increasing the COPD risk in these individuals (Smith et al., 2018).

Fgf10+/− mice showed reduced lung function similar to that in patients with heterozygous mutations in FGF10 and COPD risk. However, this phenotype may result from the smaller size of Fgf10+/− mice compared to wild-type animals (Klar et al., 2011). Fgf10 expression also occurs in murine airway smooth muscle cells (ASMCs) (Chu et al., 2021). After an injury, the transient expression of Fgf10 in ASMCs may be abnormal for adequate airway epithelial regeneration, and sustained secretion of Fgf10 by ASMCs can cause airway abnormalities resembling the defects observed in human COPD (Chu et al., 2021).

Cleft lip and palate syndrome

Nonsyndromic cleft lip with or without cleft palate (NSCL/P) are common birth defects of complex etiology, occurring in about 1/1000 live births worldwide (Dixon et al., 2011). While environmental factors influence the development of this disorder, it has been suggested that genetic causes also contribute to the formation of cleft lip and palate (CLP) (Riley et al., 2007). FGF10 is one of the members of the FGF family whose signaling pathway is important in craniofacial development and its abnormalities have been associated with clefts in the craniofacial region (Riley et al., 2007). Family-based association testing in the Philippine population with NSCL/P showed borderline significance for two markers: rs1448037 and rs1482685 in FGF10 (Riley et al., 2007). In contrast, subsequent studies conducted in the Polish population did not show any relationship between the cleft lip and palate and the rs1448037 variant in FGF10 (Mostowska et al., 2010). The variant rs10462065, located in proximity to FGF10, was recognized as a possible risk factor for CLP in the Chinese population (Yu et al., 2017). Another study performed in the Chinese population showed a correlation between rs2330542 in FGF10 and rs1946295 in TBX5, or rs7704166 in FGF10 and rs7085073 in FGFR2 that may be etiologically associated with NSCL/P. However, functional studies should be performed to clarify these statistically significant associations (Li et al., 2019). Further evidence indicating that abnormalities in FGF10 can contribute to the development of cleft palate was the identification of a deletion in the intron of FGF10 (chr5:44,347,532-44,347,538) in affected patients (Shi et al., 2009).

Mice studies support the role of FGF10 in the development of CLP in humans.

Fgf10−/− murine mutants show a complete cleft of the secondary palate caused by improper tongue attachment to the anterior palatal shelves and the middle and posterior mandible (Riley et al., 2007). SHH protein has been suggested to stimulate mesenchymal proliferation and was proposed to be a downstream target for the FGF10/FGFR2 pathway (Rice et al., 2004). Abnormal signaling of this epithelial-mesenchymal pathway leads to the development of cleft palate in mice (Rice et al., 2004). Inactivation of Shh in mouse palatal epithelium resulted in a cleavage phenotype resembling that observed in Fgfr10−/− and Fgfr2b−/− animals (Rice et al., 2004). Furthermore, the SHH activation affects the FGFR2 signaling during human craniofacial development (Raju et al., 2021). Thus, disruption of this pathway in humans may also contribute to palate defects.

Myopia

Myopia is one of the most common eye abnormalities identified in humans (Holden et al., 2016). High myopia, characterized by refractive error (RE) ≤ -6 diopters (D) or axial length (AL) ≥ 26 mm (Kempen et al., 2004; Pan, Ramamurthy & Saw, 2012), and extreme myopia, with RE ≤−10.00 D or AL ≥ 30 mm, can lead to pathological changes in the eye, and disturbing vision (Jiang et al., 2019). High myopia can cause glaucoma, retinal detachment, or macular degeneration, resulting in visual impairment (Saw et al., 2005; Jiang et al., 2019). The incidence of myopia varies between populations and gender and also depends on the patient’s age (Rose et al., 2008; Pan, Ramamurthy & Saw, 2012; Hsi et al., 2013).

The rs339501-G allele of FGF10 is a binding site for three different transcription factors and has been identified as a putative risk marker for extreme myopia in the Chinese population living in Taiwan (Hsi et al., 2013). The reporter assay showed that the G risk allele may result in a higher expression of FGF10, suggesting that increased FGF10 expression might enhance susceptibility to myopia (Hsi et al., 2013). However, a study performed in the Japanese population contradicted the findings of these studies in the Chinese population, showing that the rs339501-A is an extreme myopia risk allele (Yoshida et al., 2013). Although initial studies revealed an association of rs399501 with extreme myopia (Hsi et al., 2013), recent analyses showed that rs399501 is also significantly related to high myopia in Han Chinese (Jiang et al., 2021). Additionally, in a Japanese population, it was found that the rs12517396-C and rs10462070-A alleles in FGF10 may be associated with extreme myopia (Yoshida et al., 2013). A subsequent study in a western Chinese population confirmed the association of the rs12517396 and rs10462070 variants in extreme and high myopia (Jiang et al., 2019).

Interestingly, another variant, rs2973644, correlated with a higher risk for myopia (Sun et al., 2019), was also linked with a higher risk for COPD, as mentioned above (Ren et al., 2013). The rs2973644, together with the rs399501 and rs79002828 variants, indicated a risk of high myopia in young Chinese children (Sun et al., 2019). Furthermore, the G risk allele rs2973644 leads to an increase in FGF10 expression as opposed to the protective A allele (Sun et al., 2019).

Bronchopulmonary dysplasia

Bronchopulmonary dysplasia (BPD) is characterized by respiratory failure, lung collapse, or hypoxemia, caused by a deficiency of surfactant coating the inner surface of the lungs due to abnormal lung development during the saccular stage (Coalson, 2003)

It has been shown that activation of Toll-like receptor 2 (TLR2) or Toll-like receptor 4 (TLR4) in mice inhibits the FGF10 expression, resulting in impaired airway morphogenesis (Benjamin et al., 2007) with abnormal myofibroblast positioning at the saccular airway observed in BPD (Benjamin et al., 2007). A subsequent study showed that soluble inflammatory mediators in the tracheal fluid of earlier-born children, through NF- κB, can also inhibit the FGF10 expression resulting in defects in epithelial-mesenchymal interactions during lung development (Benjamin et al., 2010; Carver et al., 2013).

Interestingly, hyperoxia-induced neonatal lung injury was used as a mouse model of BPD to study the effect of Fgf10 deficiency in Fgf10+/− pups (Chao et al., 2017). In normoxia, no mortality was observed in either Fgf10+/+ or Fgf10+/− mice, while in hyperoxia, all Fgf10+/− mice died within 8 days, and all Fgf10+/+ mice were alive (Chao et al., 2017). Analysis of hyperoxic Fgf10+/− lungs showed increased hypoalveolarization and a lower ratio of type II alveolar epithelial cells (AECII) to total Epcam-positive cells compared to lungs in normoxia (Chao et al., 2017). Lower levels of Fgf10 transcripts lead to congenital lung defects with postnatal survival but reduced ability to cope with sublethal hypertoxic injury (Chao et al., 2017). Thus, deficiency of AECII cells caused by decreased level of FGF10 may be an additional complication in BPD patients (Chao et al., 2017). Furthermore, the lungs of Fgf10+/− mice with hyperoxia were characterized by a decreased number of blood vessels with an increase of poorly muscularized vessels (Chao et al., 2019). This may represent an additional feature of the BPD (Chao et al., 2019). However, a recent study showed that overexpression of Fgf10 and administration of rFGF10 rescued alveologenesis defects in transgenic mice (Taghizadeh et al., 2022).

Cancer

FGF10 activates intracellular signaling pathways in several cell types that can lead to cancer cell invasion and proliferation (Watson & Francavilla, 2018). Thus, abnormal regulation of FGF10 may contribute to certain forms of cancer (Itoh, 2016).

Pancreatic cancer is a common exocrine neoplasm and one of the most deadly diseases (Ndlovu et al., 2018). In a physiological state, FGF10 is involved in the formation of pancreatic epithelial cells (Bhushan et al., 2001; Ndlovu et al., 2018) and maintains undifferentiated pancreatic progenitor cells (Norgaard, Jensen & Jensen, 2003). In contrast, abnormal expression of the FGF7 and FGF10 genes in stromal cells surrounding pancreatic cancer cells have been observed (Ndlovu et al., 2018), indicating the role of FGF7 and FGF10 in cell proliferation (FGF7), migration, and invasion (FGF10) (Ndlovu et al., 2018). Moreover, ectopic expression of FGF10 can cause pancreatic hyperplasticity (Norgaard, Jensen & Jensen, 2003).

FGF10 may also be involved in a subset of human breast cancers (Stacey et al., 2008; Reintjes et al., 2013; Ghoussaini et al., 2016). Studies performed in 2008 identified rs4415084 and rs10941679 variants located approximately 274–317 kb downstream from FGF10 as likely associated with a higher risk of breast cancer (Stacey et al., 2008). While rs4415084 was subsequently excluded from breast cancer causality, the rs10941679-G allele has been associated with a 15% higher risk of estrogen-receptor-positive breast cancer (Ghoussaini et al., 2016). This variant maps to a putative enhancer interacting with the FGF10 promoter regions in breast cancer cell lines, suggesting that it could regulate the FGF10 expression (Stacey et al., 2008; Reintjes et al., 2013; Ghoussaini et al., 2016). Interestingly, FGF10 had strongly increased expression in 10% of breast cancers compared to healthy tissues (Theodorou et al., 2004).

Deregulation of FGF10-related signaling pathways was also associated with gastric cancer (Carino et al., 2021). This assumption was further supported by Wu et al., who indicated that regulatory networks involving FGF10 play an essential role in gastric cancer proliferation, migration, and invasion (Wu, Liu & Zhang, 2022).

Other diseases

Because of its role in orchestrating various developmental processes, abnormalities in FGF10 have also been described in the context of other disorders. However, the importance of FGF10 in these diseases is inconclusive and further studies are required to determine the impact of the FGF10 impairment on some defects.

Among diseases in which the role of FGF10 was not well established are conotruncal defects (CTDs), rare heart diseases with an incidence of 0.1 ‰ in live births that account for ∼25–30% of all non-syndromic congenital heart diseases (Shah et al., 2008; Zhou et al., 2020). Zhou et al. identified two rare heterozygous variants c. 29G>A and c.551G>A in FGF8 in two patients with tetralogy of Fallot and one c.68_70del variant in FGF10 in a patient with complete atrioventricular valve defect, pulmonary valve stenosis, single atrium, and single ventricle (Zhou et al., 2020). Whereas c.29G>A (p.Cys10Ter) in FGF8 and c.68_70del (p.Cys23del) in FGF10 may affect the protein function by reducing their secretion, c.551G>A (p.Arg184His) in FGF8 can negatively impact the proliferation of human cardiomyocytes, leading to CTD (Zhou et al., 2020). However, these assumptions have not been confirmed in subsequent studies.

Another condition that requires further research on FGF10 is anorectal malformation (ARM), including congenital malformations that affect the development of the distal colon (Krüger et al., 2008). Approximately 40–50% of all ARM cases have isolated ARMs, sometimes linked with malformations in the kidneys, the genitourinary system, or other systems (Stoll et al., 2007). The remaining ARM cases are associated with the spectrum of specific genetic syndromes (Draaken et al., 2012). Of note, changes in the FGFR2 gene involved in the FGF10 signaling are known to cause various forms of autosomal dominant craniosynostosis syndrome (Draaken et al., 2012). They are also associated with ARM in patients with Apert syndrome, Pfeiffer syndrome types 1 and 2, Crouzon syndrome, and Beare-Stevenson syndrome (Draaken et al., 2012).

Genetic studies have not confirmed the impact of variants in FGF10 and other genes on ARM development in the screened patients (Krüger et al., 2008; Draaken et al., 2012). However, the lack of association between ARM and FGF10 variants could be the result of a limited number of tested samples or the use of a method that was unable to detect variants in the previously unknown regulatory sequences or non-coding sequences (Draaken et al., 2012). FGF10 is likely a key factor regulating the growth in endoderm or mesenchyme and thus also stimulates the development of the genitourinary system and the anus (Yucel et al., 2004; Fairbanks et al., 2004). Future research should be extended to include additional factors within the WNT/FGF signaling pathway (Draaken et al., 2012).

Summary

FGF10 has diverse functions in organ development and their proper functioning and its alterations have been found to lead to various diseases in humans. Despite advances in understanding the pathways involved in FGF10 regulation and the discovery of new pathogenic variants in the FGF10 gene, there are still many unknowns regarding the exact role of FGF10 abnormalities in the disease etiology. Further studies of the FGF10 gene and its regulatory elements are necessary to expand our knowledge of the involvement of this gene in human diseases.

We thank Prof. Paweł Stankiewicz for helpful discussion.

Additional Information and Declarations

Competing Interests

Author Contributions

Data Availability

The authors declare there are no competing interests.

Katarzyna Bzdega conceived and designed the experiments, performed the experiments, analyzed the data, prepared figures and/or tables, authored or reviewed drafts of the article, and approved the final draft.

Justyna A. Karolak conceived and designed the experiments, analyzed the data, prepared figures and/or tables, authored or reviewed drafts of the article, and approved the final draft.

The following information was supplied regarding data availability:

This is a literature review article.

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
