# Peer review of "Phenotypic spectrum of FGF10-related disorders: a systematic review"

_PeerJ, doi:10.7717/peerj.14003_

## Round 0.1 · original submission · Major Revisions

Please address concerns of all reviewers and amend manuscript accordingly.

Reviewer 1 has suggested that you cite specific references. You are welcome to add it/them if you believe they are relevant. However, you are not required to include these citations, and if you do not include them, this will not influence my decision.

Reviewer 1 ·

Basic reporting

This is a very thorough review of an important and increasingly recognized subject. The authors have done a very admirable job in spelling out the vast field of FGF signaling in developmental biology and disease. While there have been other reviews on this subject, this particular manuscript includes more detail and perspective regarding various genetic variants and their impact on human disease.

Experimental design

The authors have done an admirable job in trying to tackle a very broad and complex subject. I would like to suggest several items that might improve the manuscript.
line 72: introduce the important role of heparin-sulfate (HS) binding earlier, as it helps differentiate the "hormonal" FGF family members and is absolutely critical for FGFR activation by the "non-hormonal" family members.
line 99: include SHH and BMP as signaling pathways regulating FGF (in particular FGF10 expression).
line 106: perhaps a little more detail on the FGF10 gene structure? And mention that the signal peptide is cleaved upon synthesis in the ER as FGF10 moves through the Golgi.
line 117: perhaps keep general FGF family effects earlier in the manuscript before discussing FGF10 specifically?
line 119: clarify the role of FGF10 in the formation vs. secretory function of the salivary and lacrimal glands.
line 124: KGF/FGF7 actually has a more pronounced effect on fluid secretion than FGF10.
line 127: role in development doesn't yet prove a role in human disease-that evidence comes later.
line 174: perhaps compare and contrast with other FGFR mutations that lead to craniofacial abnormalities via gain of function mutations?
line 204: mention that ACD is typically due to defects in FOXF1.
line 319: What cells in the pancreas express FGF10 in normal conditions/development?

Validity of the findings

While the review is extensive and comprehensive, the authors can also add bronchopulmonary dysplasia as a disease that involves loss of normal FGF10 expression. First described in humans (PMID 17071719), additional experimental studies have investigated mechanism (PMIDs: 20861353, 23558680, 27770432, 30319693, 35437594, 30566624).

Additional comments

Fantastic job! I look forward to sharing this review with my colleagues once it is published.

Reviewer 2 ·

Basic reporting

In the review article ‘Phenotypic spectrum of FGF10-related disorders: a systematic review’ the authors try to discuss the expression spectrum of FGF10 in FGF10-related disorders. The review succeeds to discuss the contents put forward in the title. The review can be considered for publishing if the review comments are addressed. Please see next sections for the comments.

Experimental design

Please see next sections for the comments.

Validity of the findings

1. English language of the paper needs to be meticulously checked before publishing anywhere. For example, line 23- correct ‘summarize’ to ‘to summarize’. I suggest the authors to go through the paper carefully to remove any grammatical/language errors.
2. Line 42, 43 is exact repetition of line 21, 22 of Abstract. I would recommend the authors to write them in a different way.
3. Line 44 to 48 is almost same as line 23 to 25. I would appreciate if the authors could edit these lines.
4. Line 89,107,155, 203, 316, 327 – References are missing. Please check for references and add wherever necessary.
5. Line 107- It cannot be 23.4 Dalton; it should be 23400 Daltons or 23.4 kDa. Please correct this.
6. There are few articles of FGF10 published in 2022 - I encourage the authors to search for the most recent articles and include them if relevant to make the review an updated one.
7. The only Figure; Figure 1 is not mentioned or discussed anywhere in the article. Please add this in the article text. I also encourage authors to add more figures if possible representing the text discussed.
8. The same applies for Tables 1 & 2- They are not included in the article text.
9. In Fig:1, what does the number of circles in the lollipop represent? For example, Cys106Phe has 4 circles and Leu73Arg has 3. Please add this information in Figure legend.
10. In Figure 1, the authors can add domain organization of the protein (signal peptide 1-37 aa, FGF domain 38-208 aa).

Reviewer 3 ·

Basic reporting

The review is well written in the contect of reported FGF10 mutants and its association with diseases.
However, I feel as a potential reader of this review, certain things can be improved.

1. Explain the fucntion of normal signalling by FGF10 in normal tissue homeostasis through a concised picture (illustration)
2. How this FGF10 signalling is disrupted by the variants in the diseased condition through an illustration will make this review to follow better.

Experimental design

No Comment

Validity of the findings

The citations are quoted till 2020, can the author update the citation to current year?

---

## Round 0.2 · Minor Revisions

Please address remaining concerns of the reviewer #2 and amend manuscript accordingly

Reviewer 1 ·

Basic reporting

The authors have improved the manuscript with the addition of clarifying information, new references, and discussions related the mechanisms regulating FGF10 and their potential connections to disease. I think this review will be of high interest to the field.

Experimental design

The review is comprehensive while still including specific mechanisms that will be of interest even to investigators familiar with the field. There appeared to be no bias in emphasizing specific hypotheses or scientific models. Organization of the review is now improved and logically presented.

Validity of the findings

I think the review does a nice job outlining what is known in the literature and where gaps remain.

Additional comments

I appreciate the authors' efforts in improving the review.

Reviewer 2 ·

Basic reporting

I thank the authors for their point by point response for my comments. I am satisfied with their response to the comments. However, I still find issues with language and requires more editorial work before it is ready to publish. Also, I find that ‘commas’ are inappropriately used at many places. Please see specific comments in the below section.

Experimental design

No comments

Validity of the findings

No comments

Additional comments

1. Line 14-16- I believe it is better to remove the portion ‘revealed in animal studies’ which makes the sentence simple and easy to understand.

2. Line 47- The authors states that ‘All manuscripts cited in this review were found and analyzed mostly from the electronic database PubMed’ but they do not mention any other database. In that case it is more appropriate to write ‘All manuscripts cited in this review were found and analyzed from the electronic database PubMed’ by omitting the word ‘mostly’.

3. Line 55 to 57- I have suggestions with the choice of words. Consider changing ‘Due to their mechanism’ to ‘Based on their mechanism’.

4. Line 74 to 77- It would be great if you add 1 or 2 sentences explaining how FGF19 family regulates glucose and lipid metabolism.

5. Line 79 to 82- I would appreciate if the authors could add few lines to understand the readers how members of the FGF11 subfamily are involved in neuronal development

6. Line 86- ‘characterized by a three domains’- please remove ‘a’.

7. Line 105- Why Spry, SEF, MKP3, BMP, and SHH are put in brackets. Are these proteins explained anywhere else? What functions do they have?

8. Line 157- Expand SNV. Please make sure that abbreviations are expanded when they are used for the first time in text. Expanded form is given only in line 176.

9. I appreciate the authors for adding a figure explaining the FGF10 signalling. However, I find inconsistency with the text in line 100, Figure 1 and Figure legend text. In the figure legend it is written- After phosphorylation of PLCγ1, calcium is released and protein kinase C (PKC) is enabled. In the article text it is written PLCγ1 leads to the activation of protein kinase C (PKC) and calcium release. From the figure one understands that PLCγ1 phosphorylation causes calcium release and PKC activation which are independent events. Please explain carefully sequence of events and make sure that uniformity exists among text, figure and figure legend.

Reviewer 3 ·

Basic reporting

No comments!

Experimental design

the review reads better now.

Validity of the findings

No comments

Additional comments

No comments

---

## Round 0.3 · accepted · Accept

Thank you for addressing the remaining critiques and revising the manuscript accordingly.